# Association of Adherence to Specific Mediterranean Diet Components and Cardiorespiratory Fitness in Young Adults

**DOI:** 10.3390/nu12030776

**Published:** 2020-03-15

**Authors:** Mª José Santi-Cano, José Pedro Novalbos-Ruiz, María Ángeles Bernal-Jiménez, María del Mar Bibiloni, Josep A. Tur, Amelia Rodriguez Martin

**Affiliations:** 1Research Group on Nutrition: Molecular, pathophysiological and social issues, University of Cádiz, Biomedical Research and Innovation Institute of Cádiz (INiBICA), 11009 Cádiz, Spain; 2Biomedicine, Biotechnology and Public Health Department, University of Cadiz, 11003 Cádiz, Spain; josepedro.novalbos@uca.es (J.P.N.-R.); amelia.rodriguez@uca.es (A.R.M.); 3Department of Nursing and Physiotherapy, University of Cádiz, Biomedical Research and Innovation Institute of Cádiz (INiBICA), 11009 Cádiz, Spain; m.angeles.bernal@inibica.es; 4Research Group on Community Nutrition & Oxidative Stress, University of the Balearic Islands, IDISBA & CIBEROBN, 07122 Palma de Mallorca, Spain; mar.bibiloni@uib.es (M.d.M.B.); pep.tur@uib.es (J.A.T.)

**Keywords:** nuts, cardiorespiratory fitness, Mediterranean diet adherence

## Abstract

Objective: Cardiorespiratory fitness (CRF) and a healthy diet may be part of an overall healthy lifestyle. The association between cardiorespiratory fitness and adherence to an overall Mediterranean Diet (MedD) pattern and specific MedD foods has been assessed. Design: Subjects completed a lifestyle survey and dietary pattern, using the validated MedD Adherence 14-item questionnaire and two self-reported 24-h dietary recalls. Participants’ height, body weight, waist circumference (WC), and CRF (maximum oxygen uptake, VO_2max,_ ml/kg/min) were measured. Setting: University of Cádiz, Spain. Subjects: A sample of young adults (*n* = 275, 22.2 ± 6.3 years). Results: Mean VO_2max_ was 43.9 mL/kg/min (SD 8.5 mL/kg/min). Most participants had healthy CRF (75.9%). The average MedD score was 6.2 points (SD 1.8 points). Participants who consumed more servings of nuts had higher VO_2max_. Those who showed low CRF performed less physical activity (PA) and had a higher body mass index (BMI) and WC compared with those classified as having healthy CRF. Nut consumption was positively associated with VO_2max_ (β = 0.320; 95% CI 2.4, 10.7; *p* < 0.002), adjusting for sex, age, smoking PA, BMI, WC, and energy intake, showing the subjects who consumed more nuts were fitter than young adults who consumed less. Conclusions: CRF is positively associated with nut consumption but not with the overall MedD pattern and all other MedD foods in the young adults. The subjects who consumed more servings of nuts were fitter than young adults who consumed less. Moreover, fitter subjects performed more PA and had a lower BMI and WC than those who had lower fitness levels.

## 1. Introduction

Cardiorespiratory fitness (CRF) is an important marker of cardiovascular health. Low levels of CRF are considered predictors of cardiovascular diseases (CVD) in adults [1] and have been linked to excess body fat and features of metabolic syndrome in young people [2,3].

High cardiorespiratory fitness, defined as meeting or exceeding the Fitnessgram^®^ standard (maximum oxygen uptake, VO_2max_) [4] is positively associated with healthier cardiovascular risk profiles, healthier body compositions, and improved insulin sensitivity [1,2,3]. Nutrition is another important modifiable risk factor associated with health. Diets usually containing high intakes of olive oil, nuts, fruits, and vegetables, such as the Mediterranean diet (MedD), can confer many health benefits, including lower risk of developing obesity, metabolic syndrome, and CVD [4].

Current scientific evidence has demonstrated the benefits of some typical foods of the Mediterranean diet, such as nuts, in the prevention of cardiometabolic diseases, overweight, obesity, and cancer. Nuts have numerous biologically active compounds, including carotenoids, polyphenols, phytosterols, vitamins, minerals, and fiber. The antioxidant and anti-inflammatory properties of these substances might act synergistically in lowering risk factors related to some age-related diseases and highlight the importance of including nuts in a healthy diet [5].

The most common edible nuts are walnuts, almonds, hazelnuts, and peanuts. These four types represent 95% of the total nut intake in the Spanish population [6]. Nuts have unique nutritional profiles and are comprised of 43–67% fat and 8–22% protein by weight. Moreover, nuts are abundant in unsaturated fatty acids (containing both monounsaturated fatty acids and polyunsaturated fatty acids) and only 4–5% of saturated fatty acids [7,8]. Therefore, the benefits are believed to outweigh the advantages.

The results of recent systematic reviews and meta-analyses show that nut consumption appeared to be associated with lower all-cause mortality and risk of cardiometabolic disease [8]. These effects have been attributed to different mechanisms such as decreases in fasting glucose [9], total cholesterol [10], LDL-cholesterol [11], weight [12], endothelial function [13], and gut microbiota modification [14]. The most commonly demonstrated effect is the reduction in LDL-cholesterol, which is similar with all common edible nuts (almonds, peanuts, hazelnuts, cashews, macadamias, pistachios, Brazil nuts, or walnuts), and has been found to be clinically significant [8].

As both cardiorespiratory fitness and nutrition may be key to youths’ current and future health, it is important to investigate whether they are associated, as these may be potential targets for interventions to promote desirable behaviors impacting youth health or becoming lifelong habits.

Cardiorespiratory fitness is positively associated with higher usual intakes of fruit, vegetables, bread, and dairy products [15,16]. However, despite these positive findings, others have found no evidence of any association [17]. One study investigated dietary patterns and cardiorespiratory fitness in adults [15]. The authors found that increasing the “meat” pattern score was negatively associated with treadmill duration. On the other hand, an increase in the “fruit-vegetable” pattern score was positively associated with treadmill duration.

Another study observed that the rise in CRF was associated with a healthier dietary pattern in adolescents [10]. Moreover, previous work investigating food group and nutrient intakes suggests that cardiorespiratory fitness and a healthy diet may be part of an overall healthy lifestyle [18]. However, studies examining the associations of CRF and MedD adherence in young adults are scarce, so further research is required to clarify associations, particularly among young adults. A better understanding of the combined associations of CRF and diet in youth is of great importance because both CRF and diet are main modifiable factors and are recommended as the keys of prevention and treatment for metabolic syndrome in youth.

It has been hypothesized that cardiorespiratory fitness may be positively associated with high adherence to the MedD, reflecting high olive oil, fruit, vegetables, and nuts consumption, and negatively associated with consumption of “treat” type foods [18]. Therefore, the purpose of this study was to assess the association between CRF and adherence to overall MedD patterns and specific MedD foods in a sample of young adults.

## 2. Methods

The present cross-sectional study was conducted from January 2014 to June 2017 and analyzed data collected from 275 university students in Cádiz, Spain. Randomly selected students at the University of Cádiz were invited to participate in the study. The subjects voluntarily accepted to participate in the study and signed an informed consent form. They were informed that it was a general screening consisting of questionnaires, and anthropometric and CRF measurements. The study subjects included 275 young adults, 100 men (36.4%), and 175 women (63.6%). The mean age of participants was 22.2 ± 6.3 years. The protocol was approved by the University of Cádiz Research Ethics Committee and met all requirements for human subject research established by the guidelines in the World Medical Association (2000) Declaration of Helsinki: Ethical Principles for Medical Research Involving Human Subjects, with notes of clarification of 2002 and 2004, and the Guidelines on the Practice of Ethics Committees Involved in Medical Research Involving Human Subjects (3rd ed., 1996; London: The Royal College of Physicians).

The survey was administered online and using a paper format. Students supplied information on their date of birth, age, sex, alcohol intake (converted to units of alcohol per week), tobacco (number cigarettes smoked per day), and physical activity (PA). Physical activity was assessed by self-reported participation in leisure time activities during the previous month. For each activity, the number of sessions per week and the average duration per session were reported. From these data, we quantified, for each subject, the frequency and duration in minutes of activity/week and minutes/day.

Food consumption was assessed using the previously validated 14-item MedD Adherence questionnaire [19] and self-reported 24-h dietary recall on two days. The total MedD score ranges from 0 to 14 points: The higher the score, the higher the degree of adherence to the MedD pattern. Each item was scored 0 or 1. One point was given for: (1) Using olive oil as the main source of culinary fat; (2) Consumption of 4 or more tablespoons (1 tablespoon = 13.5 g) of olive oil/day; (3) Consumption of 2 or more servings (1 serving = 200 g) of vegetables/day; (4) Consumption of 3 or more pieces of fruit/day; (5) Consumption of less than 1 serving (1 serving = 100 g) of red meat or hamburger or sausages/day; (6) Consumption of less than 1 serving (12 g) of animal fat, such as butter, margarine, or cream/day; (7) Consumption of less than 1 glass (100 mL) of sugar-sweetened beverages/day; (8) Consumption of 7 or more glasses (100 mL) of red wine/week; (9) Consumption of 3 or more servings (1 serving = 150 g) of pulses/week; (10) Consumption of 3 or more servings (1 serving = 150 g) of fish/week; (11) Consumption of less than two commercial pastries/week; (12) Consumption of 3 or more servings (1 serving = 30 g) of tree nuts/week; (13) Preferring white meat over red meat; (14) Consumption of ‘‘sofrito’’ (a sauce made with tomato, onion, garlic or leek simmered with olive oil) 2 or more times/week. DIAL program (Alce Ingenieria Madrid, Spain 2008) was used for the 24 h questionnaire assessment. The questionnaires and measurements were performed to each participant within the same week.

Height, weight, and waist circumference (WC) were measured, in the morning fasting, twice at a 5 min interval. The mean of the two measurements was used. Height was measured using a calibrated portable stadiometer (Seca 213, Germany) and weight (in light clothing and without shoes) measured using a calibrated scale (HD-352 Tanita Corporation, Japan) by trained research assistants in which reliability was previously checked. Body mass index (BMI) was calculated as kg/m^2^ and then categorized into four levels; thinness (< 8.5 kg/m^2^), normal-weight (18.5−24.9 kg/m^2^), overweight (25−29.9 kg/m^2^), and obese (≥ 30 kg/m^2^). Waist circumference (WC) was measured to the nearest 0.1 cm using a non-extensible tape, midway between the lower ribs and the upper iliac crest in a standing position. The cut-off points used for men and women were 94 cm and 80 cm, respectively [20].

Léger et al.’s 20 m multistage fitness test was used to determine cardiorespiratory fitness [21]. All study personnel were trained before data collection. The 20 m multistage fitness test was performed on a sports court at the participants’ university campus. Participants were instructed to run between two parallel lines of cones, 20 m apart, pacing themselves by the pre-recorded beeps emitted from a stereo system. The test began with a pace of 8.5 km/h, increasing by 0.5 km/h every minute (or stage). If a participant failed to reach the line of cones before the next beep sounded they were warned. If two consecutive beeps were missed, the participant was withdrawn from the test and their last stage and lap were recorded. The 20 m multistage fitness test results were used to determine VO_2max_ relative to body mass (ml/kg per min) for each participant using the predictive equations of Léger et al. [15]. Participants were additionally categorized into low and high cardiorespiratory fitness using the Fitnessgram^®^ standards cut-offs (criterion VO_2max_ of 42 mL/kg per min and 35 mL/kg per min for young adults men and women, respectively) [4]. These values were in the 25^th^ percentile for VO_2max_ in the sample.

All analyses were performed via Statistical Package for Social Sciences (SPSS) software, version 21.0 for Windows. Continuous variables were checked for normality and are presented as mean and standard deviation (SD), with categorical variables as frequencies. According to the 14-item MedD questionnaire score, subjects were grouped as follows: Score 0–5, the lowest adherence to the MedD; score 6–9, medium adherence; score ≥10, the highest adherence to the MedD. Both the Student t-test and Mann–Whitney test were adopted to compare continuous variables across sex and CRF groups, and the Chi-square/Fisher test used to compare categorical variables. The percentage of fulfillment of each specific item of the 14-item MedD questionnaire was compared according to the VO_2max_ low and high groups by using a Chi-squared test. Multivariable linear regression analyses were undertaken with VO_2max_ as the dependent variable, while each of the items from the 14-items MedD questionnaire were the independent variables in regression modeling. The model was adjusted for sex, age, smoke, PA, BMI, WC, and kcal/d. A two-sided *p* < 0.05 was considered statistically significant.

## 3. Results

The participant characteristics are presented in Table 1. The study subjects included 275 young adults, 100 men (36.4%), and 175 women (63.6%). The mean age of participants was 22.2 ± 6.3 years. Most participants were normal weight (66.5%), 3.4% of subjects were classified as having a low BMI, 20.5% were overweight, and 9.5% were obese. The mean values of WC were 85.9 ± 12.4 in men and 75.4 ± 10.8 in women. Men performed more physical activity than women (*p* = 0.002). Mean VO_2max_ was 43.9 ± 8.5 mL/kg/min. Most participants had a healthy VO_2max_ (75.9%). Higher levels of cardiorespiratory fitness were seen in men compared with women.

The average MedD score was 6.2 ± 1.8. MedD medium adherence was the most frequent, at 62.8%, low, at 33.6%, and high, at 3.6%. Women showed significantly lower ‘Fruit’ and ‘Nuts’ consumption than the men (24.8% vs. 10.9%, *p*< 0.004; 20.6% vs. 9.5%, *p* < 0.018, respectively). Men showed higher energy intake than women, but nutrient distribution was similar (Table 2).

Participants who consumed more servings of nuts or “sofrito” had a higher VO_2max_ (Figure 1). Subjects who had low VO_2max_ performed less physical activity and they had higher BMI and WC compared with those classified as healthy VO_2max_ (Table 3). MedD adherence and components, as well as energy and nutrient intake, were not influenced by VO_2max_ (Table 4).

Continuous VO_2max_ model (linear regression): The association between VO_2max_ and nut consumption for the total sample was a positive linear association (β = 0.320; 95% CI 2.4, 10.7; *p* < 0.002), adjusting for sex, age, smoke, PA, BMI, WC, and Kcal/d, indicating that with increasing fitness, young adults reported consuming more nuts (Table 5). Only MedD item 12 was significantly associated with VO_2max_.

## 4. Discussion

To the best of our knowledge, this is the first study to examine the positive association between specific MedD components and cardiorespiratory fitness in young adults, and the most recent study to measure cardiorespiratory fitness in Spanish university students.

Although MedD adherence was medium in the total sample (6.2 ± 1.8 points) (Table 4), without statistical differences between men and women, the study suggests young adults with higher nut intakes showed higher VO_2max_ values.

A positive association between CRF and a healthy diet was found in a European study [18] and no evidence of any relationship was observed in an American study [15]. European adolescents considered as having high fitness were found to have lower mean intakes of beverages with added sugar than those classified as having low CRF [18]. They found comparable figures to ours for CRF in men and women, respectively (46.2 mL/kg/min vs 38.4 mL/kg/min). In agreement with these findings, our results showed that healthier patterns (MedD item 12, ≥3 servings of tree nuts/week and item 14, ≥ 2 servings of “sofrito”/week) were associated with increased cardiorespiratory fitness. Moreover, women showed significantly lower ‘Fruit’ and ‘Nuts’ consumption than the men, although the prevalence of healthy CRF was similar in both sexes. Previous work found in adolescents that a “fruit and vegetables” dietary pattern was positively associated with VO_2max_ [16]. In a recent study conducted in university students, the authors observed that a high level of adherence to the MedD was associated with a high level of CRF, although the prevalence of high adherence to MedD was low [22]. They found MedD scores of 7.2 ± 1.9 points in men and 6.9 ± 2 points in women, slightly higher than ours; 6.2 in both sexes.

The low MedD score found in the current study (6.2 ± 1.8 points) is not surprising and may have transcendence in public health. Viscogliosi et al. [23] found similar mean values and an overall low level of adherence to the MedD (score, 6.6 ± 2.4) in a population of subjects, with a mean age of 59.8 ± 10.2 years old, referred for evaluation of cardiovascular risk factors. These findings may suggest that dietary pattern changes toward modern Western diets are occurring, even in Mediterranean countries, indicating dietary globalization in Western countries. Currently, a diet rich in meat, processed foods, and sweets have become more common at a population level, mainly in young subjects.

Nevertheless, our results suggest that the frequency of consumption of nuts may exert a beneficial role on CRF, regardless of the overall MedD pattern. Despite their high energy density, nuts do not contribute to weight gain, changes in waist circumference, or obesity, perhaps due to their satiating effects and increased fecal energy losses [24]. Nuts contain bioactive and health-promoting components. They are highly nutritious and have nutrients that are recognized for their role in reducing CVD risk. This may be due to the favorable lipid profile and low-glycemic nature of nuts. Other evidence suggests that increased consumption of nuts increases antioxidant defenses and reduces inflammation [25].

We determined dietary energy, and it was similar in low and high CRF groups, but fitter students had a low BMI and WC as compared to those who were less fit. We found that fitter subjects were consuming healthier foods (nuts). The least fit subjects might eat “healthy” foods at a lower frequency because they lack nutrition knowledge. It is known that nutrition knowledge influences people’s dietary habits, promoting healthier choices, independently from other less-modifiable risk factors such as socioeconomic position [26]. In addition, the positive associations between adherence to specific MedD components in the present study (nuts, MedD item 12^th^) and CRF may be due to the effects of the healthy lifestyle (Table 5). However, randomized clinical trials are needed to confirm the influence of the adherence to specific MedD components on CRF and this could be a future direction for this work.

Maintaining or improving fitness is likely to counteract some of the adverse effects of fat gain. In multiple linear regressions, nut consumption was significantly associated with CRF. We controlled for sex, age, smoke, PA, weight status, WC, and energy intake in our model, and the positive association remained (Table 5). Physical activity may explain in part the association between nuts intake and CRF. The subjects with higher CRF are more active and more conscious of the importance of health, therefore they consume healthy foods. Recent work found that physical activity and eating behavior may be related through a common neurocognitive pathway. The authors examined the effect of moderate to vigorous exercise on the neural response (monitored using electroencephalogram) to pictures of food in normal-weight and obese women and suggested that physical activity might decrease neurologically determined food motivation [27]. Indeed, different lines of research suggest a positive influence of physical activity on the self-regulation of eating behavior by enhancing the sensibility of the physiological satiety signaling system [28]. Our results show that healthy CRF performed more PA but they did not intake more energy per day. Although we did not find a relationship between CRF and alcohol or tobacco, the healthy CRF group trended to show lower smoking habits.

The current work has measured cardiorespiratory fitness in young adults using an internationally comparable method. Concerning European adolescents [29] and young adults of a similar age [24], the sample completed more stages in the test and therefore demonstrated a higher VO_2max_ (40.6 mL/kg/min, 37.2 mL/kg/min and 43.9 mL/kg/min, respectively) but a lower prevalence of those classified as having high CRF in young adults, as compared to European adolescents (75.9% vs. 88%) [29]; although, a higher prevalence of healthy CRF was observed in adolescents (75.9% vs. 40.4%) [30]. The higher levels of CRF seen in the present study are likely related to the characteristics of our sample, which was selected among university students. This might be the reason for the high proportion of individuals that performed PA and were classified as having high CRF.

A greater level of habitual physical activity could increase CRF and enhance energy expenditure, thereby reducing body weight. In the present analysis, self-reported physical activity was lower and BMI was higher in the low group of CRF. These findings are in line with evidence from randomized controlled trials that aerobic exercise of moderate or vigorous intensity performed frequently over weeks or months significantly increases CRF in men and women [31].

Previous studies have shown that a lower CRF is associated with increased body weight [32,33]. Whether low CRF predisposes to obesity or vice versa is unknown. The variability in CRF can be partially attributed to heritable factors [34]. Nonetheless, to date, no specific genetic variants have been identified that can explain CRF levels.

Our study has some limitations. First, because of the cross-sectional design, we cannot prove causal relationships. Second, we evaluated nut intake through a self-reported questionnaire. This might cause some recall bias that could have potential effects on the study outcomes. Nevertheless, participant responses were anonymous; as a result, they had no reason to misreport. Third, we evaluated the overall intake of nuts; we were not able to analyze specific types of nuts or commercial preparation. It is important to emphasize the recommendation to consume nuts without sugar, salt, or additives and as a substitute for other energy-dense snacks that lack nutritional value to facilitate beneficial changes in dietary habits. Fourth, the sample of the cohort was comprised of health-conscious university students. Fifth, although CRF is considered an objective measure of habitual physical activity, CRF may be influenced by other factors, such as environmental (heat and humidity) or genetic causes, that could limit the ability to change CRF.

## 5. Conclusions

To our knowledge, the current study is the first one to examine the positive association between nut consumption and cardiorespiratory fitness in young adults, with no association observed between MedD Scores and cardiorespiratory fitness. The subjects who consumed more servings of nuts were fitter than the young adults who had fewer servings of these foods. Moreover, fitter subjects performed more PA and had lower BMI and WC than those who were less fit. Further longitudinal investigations are needed to confirm these associations about health outcomes and to assess whether these associations may be reversible and whether the consumption of a MedD may increase CRF at a population level.

## Figures and Tables

**Figure 1 nutrients-12-00776-f001:**
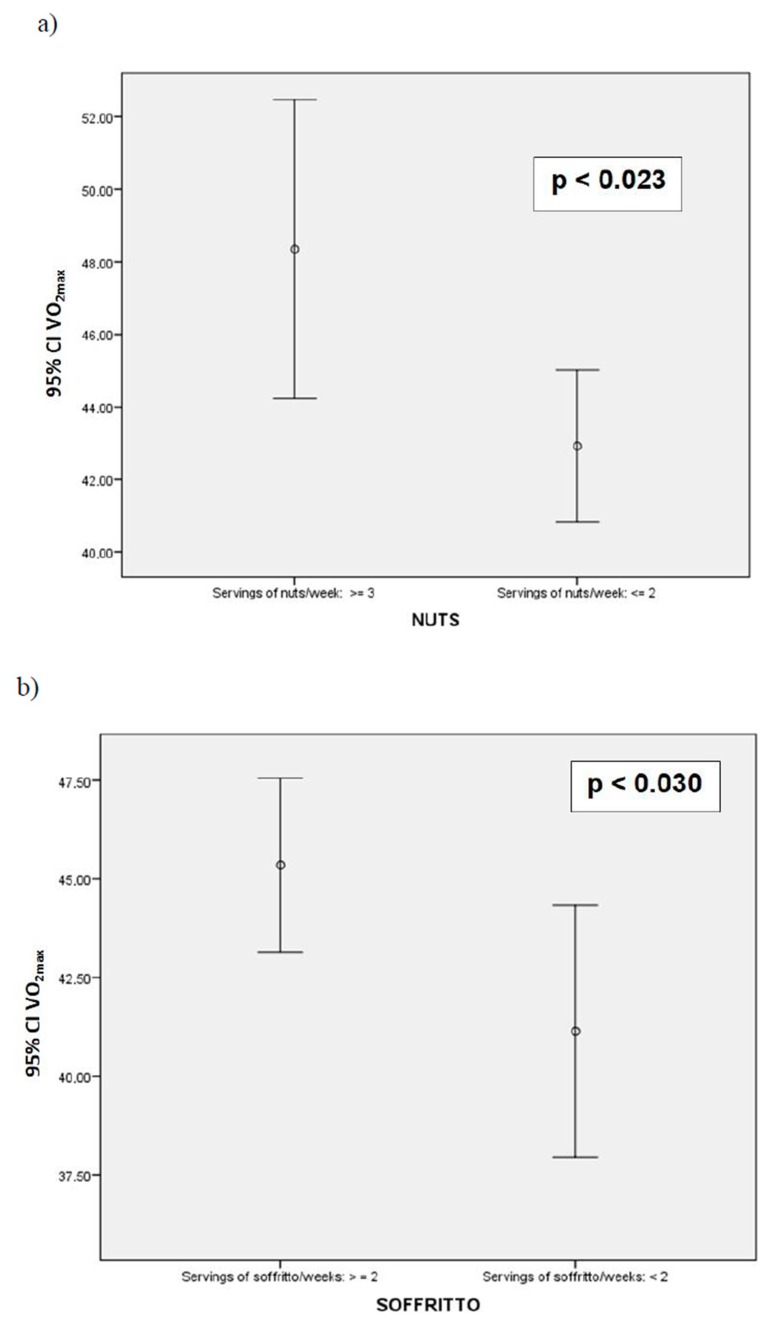
Cardiorespiratory fitness (VO_2max_) according to servings of nuts (**a**) and sauce with tomato and olive oil (**b**) consumption (Mediterranean Diet items: 12 and 14, respectively). Student’s t-test.

**Table 1 nutrients-12-00776-t001:** Characteristics of the participants according to sex.

	Total (*n* = 275)		Men (*n* = 100)		Women (*n* = 175)		*p*
	Mean	SD	Mean	SD	Mean	SD	
Age (year) *	22.2	6.3	23.0	6.7	21.7	6.0	0.004
Alcohol (units/week) *	2.6	4.1	2.9	4.2	2.3	4.1	0.118
Tobacco *	1.7	3.9	2.7	5.3	1.1	2.7	0.145
Smoking (cig./day) ^†^	15.5		15.5		15.6		0.560
PA (min/dau) *	63.2	52.0	75.9	52.4	53.1	49.6	0.002
Maximum oxygen uptake, VO_2max_ (ml/kg/min)	43.9	8.5	46.4	8.9	39.7	6.0	0.000
Healthy CRF (%) ^†^	75.9		74.1		78.8		0.797
BMI (kg/m^2^) *	24.4	6.0	25.7	7.9	23.5	4.0	0.000
Low BMI (%) ^‡^	3.4		2.9		3.7		0.006
Normal BMI (%) ^‡^	66.5		54		73		0.006
Overweight (%) ^‡^	20.5		30.4		14.3		0.006
Obesity (%) ^‡^	9.5		11.8		8.1		0.006
WC (cm) *	79.5	12.5	85.9	12.4	75.4	10.8	0.000
High WC (%) ^†^	23.3		22.0		24.1		0.763

Alcohol: Units of alcohol per week. Tobacco: Number cigarettes smoked per day. CRF: Cardiorespiratory fitness. PA: Physical activity, minutes/day. BMI: Body mass index (low: < 18.5, normal: 18.5−24.9, overweight: 25−29.9, obesity: ≥ 30). WC (waist circumference) cutoffs (high: ≥ 94 in men, ≥ 80 in women). * U Mann–Whitney. ^†^ Chi-square. ^‡^ Chi-square (Low BMI, normal BMI, overweight, obesity *p* = 0.006).

**Table 2 nutrients-12-00776-t002:** Fulfillment of the specific Mediterranean diet adherence items and 24-h dietary recall according to sex.

	Total Mean ± SD		Men Mean ± SD		Women Mean± SD		*p*
Mediterranean Diet (MedD) score (mean ± SD)	6.2	1.8	6.2	2.0	6.2	1.7	0.901
MDA low ≤5 (%)	33.6		34.5		32.9		0.741
MDA medium 6−9 (%)	62.8		60.9		64.1		0.741
MDA high ≥10 (%)	3.6		4.5		3.0		0.741
1. Using olive oil as the main culinary fat (%)	94.9		96.4		94.0		0.419
2. ≥4 Table spoons olive oil/day (%)	13.4		10.0		15.7		0.209
3. ≥2 Servings of vegetables/day (%)	38.7		31.8		43.3		0.059
4. ≥3 Pieces of fruit/day (%)	16.4		24.8		10.9		0.004
5. <1 Serving of red or processed meat/day (%)	43.3		37.6		47.0		0.137
6. <1 Serving of butter, margarine or cream/day (%)	85.3		84.4		86.0		0.730
7. <1 Glass of sugar-sweetened beverages/day (%)	72.1		67.3		75.3		0.171
8. Moderate wine consumption (%)	1.1		0.9		1.2		1.0
9. ≥3 Servings of pulses/week (%)	21.5		20.2		22.4		0.764
10. ≥3 Servings of fish/week (%)	13.4		13.6		13.3		1.0
11. <2 Commercial pastries/week (%)	75.1		77.3		73.7		0.571
12. ≥3 Servings of tree nuts/week (%)	14.0		20.6		9.5		0.018
13. Preferring white meat over red meat (%)	74.7		74.5		74.9		1.0
14. ≥2 “Sofrito”/week (%)	65.7		70.0		62.9		0.246
Energy (Kcal/d) *	1886	645	2140	715	1719	535	0.000
% Protein	17.4	4.1	17.2	4.2	17.6	4.1	0.421
% Total fat	40.4	7.8	39.6	7.7	40.9	7.8	0.208
% SFA (Saturated fatty acids)	13.3	3.6	12.8	3.6	13.6	3.5	0.121
% MUFA (monounsaturated fatty acids)	18.4	4.8	18.4	4.8	18.4	4.7	0.933
% PUFA (polyunsaturated fatty acids) *	5.3	1.8	5.2	1.5	5.4	2.0	0.298
Cholesterol (mg/day) *	283	155	294	139	275	165	0.143
% Carbohydrates	41.9	8.8	42.8	8.5	41.3	8.9	0.203
Fiber (g/day) *	18.3	9.2	20.9	9.8	16.7	8.4	0.001

MDA: Mediterranean diet adherence. * U Mann–Whitney.

**Table 3 nutrients-12-00776-t003:** Characteristics of the participants according to VO_2max_ (low and high).

	Total (*n* = 258)		Low (*n* = 63)		High (*n* = 195)		*p*
	Mean	SD	Mean	SD	Mean	SD	
Age (y) *	21.9	5.8	24.3	8.7	21.11	4.2	0.017
Alcohol (units/week) *	2.2	3.2	2.0	2.7	2.3	3.4	0.863
Tobacco (cig./day) *	3.6	5.6	5.0	8.6	3.3	4.9	0.974
Smoking (%)	20.7		19.0		21.2		1.0
PA (min/d) *	78	58	54	66	87	52	0.002
VO_2max_ (ml/kg/min)	43.9	8.5	33.7	5.1	47.1	6.6	0.000
BMI (Kg/m^2^) *	23.7	3.8	26.9	5.0	22.7	2.8	0.000
Low BMI <18.5 (%) ^†^	3.4		0.0		4.5		0.000
Normal BMI 18.5−24.9 (%) ^†^	71.3		38.1		81.8		0.000
Overweight 25−29.9 (%) ^†^	14.9		28.6		10.6		0.000
Obesity >30 (%) ^†^	10.3		33.3		3.1		0.000
WC (cm) *	79.6	11.7	87.5	12.8	77.0	10.2	0.001
High WC (%)	15.3		33.3		9.4		0.014

Alcohol: units of alcohol per week. Tobacco: number cigarettes smoked per day. PA: Physical activity, minutes/day. BMI: Body mass index (low: < 18.5, normal: 18.5−24.9, overweight: 25−29.9, obesity: ≥ 30). WC (waist circumference) cutoffs (High: ≥ 94 in men, ≥ 80 in women). * U Mann–Whitney. ^†^ Chi-square. (Low BMI, normal BMI, overweight, obesity *p* = 0.006).

**Table 4 nutrients-12-00776-t004:** Mediterranean diet adherence and 24-h dietary recall (two days) data by VO_2max_ (low and healthy).

	Total (*n* = 258) Mean ± SD		Low (*n* = 63) Mean ± SD		Healthy (*n* = 195) Mean ± SD		*p*
MedD score *	6.2	1.8	6.1	2.2	6.4	1.8	0.319
MDA low ≤5 (%)	27.6		42.9		22.7		
MDA medium 6−9 (%)	69.0		52.4		74.2		0.167
MDA high ≥10 (%)	3.4		4.8		3.0		
1. Using olive oil as the main culinary fat (%)	94.3		100		92.4		0.330
2. ≥4 Tablespoons olive oil/day (%)	13.8		4.8		16.7		0.279
3. ≥2 Servings of vegetables/day (%)	43.0		47.6		41.4		0.800
4. ≥3 Pieces of fruit/day (%)	27.6		28.6		27.3		1.0
5. <1 Serving of red or processed meat/day (%)	42.5		38.1		43.9		0.801
6. <1 Serving of butter, margarine or cream/day (%)	82.8		81.0		83.3		0.751
7. <1 Glass of sugar-sweetened beverages/day (%)	73.6		61.9		77.3		0.255
8. Moderate wine consumption (%)	1.1		0.0		1.5		1.0
9. ≥3 Servings of pulses/week (%)	20.0		25.0		18.5		0.533
10. ≥3 Servings of fish/week (%)	11.5		14.3		10.6		0.699
11. <2 Commercial pastries/week (%)	73.6		72.7		76.2		1.0
12. ≥3 Servings of tree nuts/week (%)	19.3		9.5		22.6		0.336
13. Preferring white meat over red meat (%)	72.4		76.2		71.2		0.783
14. ≥2 “Sofrito”/week (%)	66.7		52.4		71.2		0.121
Energy (Kcal/day) *	2142	788	1914	880	2223	746	0.088
% Protein	16.6	3.6	15.7	3.9	16.9	3.5	0.226
% Total fat	39.9	6.8	39.8	8.1	40.0	6.4	0.913
% AGS (Saturated fatty acids)	13.0	3.5	13.2	4.3	12.9	3.2	0.743
% AGM (monounsaturated fatty acids)	18.4	4.4	18.3	4.9	18.5	4.2	0.447
% AGP (polyunsaturated fatty acids) *	5.3	1.7	5.1	1.6	5.4	1.8	0.421
Cholesterol (mg/day) *	292	147	246	142	309	147	0.150
% Carbohydrates	43.3	8.3	44.4	10.1	42.9	7.7	0.531
Fiber (g/day)	21.9	9.9	20.0	11.2	22.5	9.4	0.263

VO_2max_ cutoffs: 42 mL/kg/min for men and 35 mL/kg/min for women. MDA: Mediterranean diet adherence. * U Mann–Whitney.

**Table 5 nutrients-12-00776-t005:** Standardized beta coefficients from linear multiple regressions with cardiorespiratory fitness as a dependent variable.

	B	95% CI	*p*
Item 12. ≥ Servings of tree nuts/week (%)	0.320	2.4, 10.7	0.002
Sex	−0.590	−14.6, −6.7	0.000
Age	−0.299	−0.7, −0.1	0.003
Smoke	0.126	−1.2, 6.3	0.187
PA	0.061	−0.0, 0.0	0.532
BMI	−0.443	−8.3, −2.2	0.001
WC	−0.182	−0.3, 0.0	0.169
Kcal/d	−0.041	−0.0, 0.0	0.687

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
