# Peer review of "Association of Adherence to Specific Mediterranean Diet Components and Cardiorespiratory Fitness in Young Adults"

_nutrients, 2020, doi:10.3390/nu12030776_

Round 1

Reviewer 1 Report

The paper regards interesting topics related to current nutrition trends. The results obtained are valuable and new information for people who want to adhere and implement a proper diet and healthy lifestyle. Despite the positive assessment indicated, in my opinion some parts of the work need to be corrected.

Please find details of review below:

Title of manuscript

In my opinion, the title of the paper was not chosen correctly. Various factors were examined in the work, including nuts, which were supposed to affect CRF. The paper actually confirmed the positive relationship between these two factors but this was one of the conclusions. I propose to change the title to properly correspond to the goal that the authors specified at work.

Introduction

I propose to extend the introduction to information on the nutritional properties of nuts, that is, an indication of why nuts could be a valuable component of the diet acting preventively and at the same time improving the state of our body (composition of fatty acids and other valuable ingredients).

Methods:

I have some doubts about the quality of the sample. Because at the moment we have 2020 and the survey was conducted 6 years ago. Can we therefore consider the achievement of this work as present and credible? Please refer to this suggestion why the results have not been published for so long, is there any explanation for this?

Page 2 line 44

Please, explain what do you mean “a major source of fat”??

Most early observational data point to a reduction in cardiovascular disease risk with moderate, isocaloric, nut consumption as a major source of fat

Page 2 line 45

In my opinion this sentence is very controversial:

One early evaluation of observational studies showed that substituting nuts for a serving of carbohydrates or saturated fats reduced blood lipids, inflammatory and oxidants mediators, body weight, as well as the risk for cardiovascular disease(6). ??

Page 3 line 143

The study subjects included 275 young 142 adults, 100 men (36.4%) and 175 women (63.6%). The mean age of participants was 22.2±6.3 years. – this information should be moved above, where the authors describe the study (page 2 line 76)

Page 9 line 245

Please provide that this information regards the data in Table 5.

Table 3

There is no need to use a legend. Please remove and write as follows: Legend the same like in Table 1.

Should be correct:

Page 2 line 77 wereinvited - were invited

Page 3 line 116 todetermine -  to determine

Page 10 line 288 relationto  - relations to

Author Response

Angellucky Iv

Assistant Editor

Nutrients Editorial Office

                                                                                                      February 20, 2020

Dear Editor, 

Thank you for reading our manuscript titled “Association between a healthy dietary habit such as nuts consumption and cardiorespiratory fitness in young adults” (Manuscript ID: 719953) and reviewing it, which has helped us improve it. We revised the manuscript and all suggested changes have taken place.

Please, find below our point-by-point answers to reviewers´ comments (blue color font). Changes made in the revised manuscript are in red letters and the words deleted highlighted in yellow.

Reviewer #1:

“The paper regards interesting topics related to current nutrition trends. The results obtained are valuable and new information for people who want to adhere and implement a proper diet and healthy lifestyle”.

We thank the reviewer for these comments.

“Despite the positive assessment indicated, in my opinion some parts of the work need to be corrected.

Please find details of review below:

Title of manuscript

In my opinion, the title of the paper was not chosen correctly. Various factors were examined in the work, including nuts, which were supposed to affect CRF. The paper actually confirmed the positive relationship between these two factors but this was one of the conclusions. I propose to change the title to properly correspond to the goal that the authors specified at work”.

In the revised manuscript, we have changed the title to properly correspond to the goal of the work.

“Introduction

I propose to extend the introduction to information on the nutritional properties of nuts, that is, an indication of why nuts could be a valuable component of the diet acting preventively and at the same time improving the state of our body (composition of fatty acids and other valuable ingredients)”.

In the revised manuscript, we have added, in the Introduction section, information on the nutritional properties of nuts and three references (Ref. nº 5-7) to support the new text (Page 2, line 51-61).

“Methods:

I have some doubts about the quality of the sample. Because at the moment we have 2020 and the survey was conducted 6 years ago. Can we therefore consider the achievement of this work as present and credible? Please refer to this suggestion why the results have not been published for so long, is there any explanation for this?”

In the revised manuscript, we have specified the start and end date of the study not only the recruitment date that had been indicated in the previous version (Page 3, line 100). The work began with the recruitment of the participants and then the questionnaires and measurements were carried out. The cardiorespiratory fitness test was performed individually, by trained personnel, on a sports court. Telephone calls were made when participants were absent from any scheduled appointments, to encourage them to participate and a new date was proposed. This extended the duration of the study.

“Page 2 line 44

Please, explain what do you mean “a major source of fat”??

Most early observational data point to a reduction in cardiovascular disease risk with moderate, isocaloric, nut consumption as a major source of fat

Page 2 line 45

In my opinion this sentence is very controversial:

One early evaluation of observational studies showed that substituting nuts for a serving of carbohydrates or saturated fats reduced blood lipids, inflammatory and oxidants mediators, body weight, as well as the risk for cardiovascular disease(6). ??”

In the revised manuscript, we have replaced these sentences with others and we added one updated references (Ref. nº 8, A review of meta-analyses) to support the new text (Page 2, line 62-68).

“Page 3 line 143

The study subjects included 275 young 142 adults, 100 men (36.4%) and 175 women (63.6%). The mean age of participants was 22.2±6.3 years. – this information should be moved above, where the authors describe the study (page 2 line 76)”

In the revised manuscript, we have moved the text above where we describe the study (Page 3, line 103-105).

“Page 9 line 245

Please provide that this information regards the data in Table 5”.

In the revised manuscript, we have provided that this information is shown in Table 5 (Page 9, line 272).

“Table 3

There is no need to use a legend. Please remove and write as follows: Legend the same like in Table 1”.

In the revised manuscript, we have removed the legend and we have written “Legend: The same like in Table 1” (Page 7, line 217).

“Should be correct:

Page 2 line 77 wereinvited - were invited

Page 3 line 116 todetermine -  to determine

Page 10 line 288 relationto  - relations to”

In the revised manuscript, we have corrected these words:

Page 3, line 102.

Page 4, line 143.

Page10, line 322. The expression “in relation to” has been replaced by “about”.

We hope that the manuscript is now acceptable for publication in Nutrients.

Yours sincerely,

Mª José Santi, MD, PhD.

Reviewer 2 Report

Nutrients                                                                                                        Feb 12, 2020

This study describes an association with nut consumption and CRF.  This manuscript is interesting and yet leaves the reader with several key questions.  Namely, the methods state that “consumption of 3 or more servings (1 serving = 30 g) of treenuts/week”.  This is vague.  What type of treenuts?  Is it presumed that all treenuts have the same nutritional value, fat content, ect…are these roasted, un-salted, i.e. processed vs. unprocessed?  This in my opinion needs to be flushed out in much greater detail, as certainly not all commercially available nuts possess the same quality, in fact heavily salted/sweet peanuts may possess additives and preservatives that may do more harm than good. 

Additionally, the manuscript could have benefited from further editing as there are several sentences which require some review.

Limitations section was missing (referencing recall bias as a potential source of error), and although the topic appears novel, it is hard to believe that the topic has not been studied before.

Confidence on the outcome generally is shaky due to the above addressed concerns.

Comments:

MedD was listed in some sections of the manuscript, however in other areas the acronym is not used (see Line 49, 65, 93, ect…)

Some minor typographical mistakes:

Line 62, should read “adolescents” add and s.

Line 77, space between “wereinvited”.

Lines 10-111, awkward sentence, were should read “was”;  (in light clothing w/out shoes) I would place in brackets;  “of” thinness instead of “as”…

Line 116, separate “todetermine”.

Line 137-140, awkward sentence, should read “as the dependent variable” insert the, “Items” should not be capitalized, insert “the” before the 14-item MedD questionnaire.

Line 145, check ALL % throughout the text, in some cases spacing after #, in others not.

Line 230, should read in young “subjects”.  Add an s on subjects.

Line 288, in “relationto” should read as two words.

Ques:

Were participants blinded to the study, i.e. is there a chance that they could have over-embellished the “healthy dietary habits”?

Is there a downfall to possibly eating too many nuts, (i.e. mold, pesticides), along these lines was there further qualification of the tree nuts to organic Yes/No.

Give above comments, the introduction and discussion should be reviewed.

Author Response

Angellucky Iv

Assistant Editor

Nutrients Editorial Office

                                                                                                      February 20, 2020

Dear Editor, 

Thank you for reading our manuscript titled “Association between a healthy dietary habit such as nuts consumption and cardiorespiratory fitness in young adults” (Manuscript ID: 719953) and reviewing it, which has helped us improve it. We revised the manuscript and all suggested changes have taken place.

Please, find below our point-by-point answers to reviewers´ comments (blue color font). Changes made in the revised manuscript are in red letters and the words deleted highlighted in yellow.

REVIEWER 2

This study describes an association with nut consumption and CRF.  This manuscript is interesting and yet leaves the reader with several key questions.  Namely, the methods state that “consumption of 3 or more servings (1 serving = 30 g) of treenuts/week”.  This is vague.  What type of treenuts?  Is it presumed that all treenuts have the same nutritional value, fat content, ect…are these roasted, un-salted, i.e. processed vs. unprocessed?  This in my opinion needs to be flushed out in much greater detail, as certainly not all commercially available nuts possess the same quality, in fact heavily salted/sweet peanuts may possess additives and preservatives that may do more harm than good. 

In our study nuts consumption was assessed using the previously validated 14-item Mediterranean Diet Adherence questionnaire. This questionnaire has been used extensively in scientific studies. We assessed overall nut intake. Observational studies, systematic reviews and meta-analysis consulted evaluated the servings of nuts per week without specifying the elaboration. Walnuts, almonds, hazelnuts and peanuts represent 95% of the total nut intake in the Spanish population (Bes-Rastrollo et al. Obesity 2007; 15:107-16). Certainly the way of elaboration can influence the caloric content of nuts and we have added this issue in the limitations, in the discussion section (Page 10, line 308-310).

“Additionally, the manuscript could have benefited from further editing as there are several sentences which require some review”.

The revised manuscript has been edited.

“Limitations section was missing (referencing recall bias as a potential source of error), and although the topic appears novel, it is hard to believe that the topic has not been studied before.

Confidence on the outcome generally is shaky due to the above addressed concerns”.

In the revised manuscript we have included the Limitations section in the discussion section (Page 10, line 308-315).

In the revised manuscript we have added updated text on the topic of the relationship between cardiorespiratory fitness and adherence to Mediterranean diet supported by a new reference (Page 9, line 245-248, Ref. nº 16). We have also searched in PubMed and we have not found studies on the association between nut consumption and cardiorespiratory fitness.

“Comments:

MedD was listed in some sections of the manuscript, however in other areas the acronym is not used (see Line 49, 65, 93, ect…)”

In the revised manuscript, we have used the acronym throughout the text.

Some minor typographical mistakes:

Line 62, should read “adolescents” add and s.

Line 77, space between “wereinvited”.

Lines 10-111, awkward sentence, were should read “was”;  (in light clothing w/out shoes) I would place in brackets;  “of” thinness instead of “as”…

Line 116, separate “todetermine”.

Line 137-140, awkward sentence, should read “as the dependent variable” insert the, “Items” should not be capitalized, insert “the” before the 14-item MedD questionnaire.

Line 145, check ALL % throughout the text, in some cases spacing after #, in others not.

Line 230, should read in young “subjects”.  Add an s on subjects.

Line 288, in “relationto” should read as two words.

In the revised manuscript, we have corrected all these typographical mistakes.

Page 2, line 87.

Page 3, line 102.

Page 3, line 134-138.

Page 4, line 143.

Page 4, line 165.

All % throughout the text have been checked.

Page 9, line 256.

Page 10, line 322.

“Ques:

Were participants blinded to the study, i.e. is there a chance that they could have over-embellished the “healthy dietary habits”?”

The subjects voluntarily accepted to participate in the study and signed an informed consent form. They were health-conscious university students. We evaluated nut intake through a self-reported dietary questionnaire. This might cause some recall bias. All studies that include dietary questionnaires may have this limitation. We have added this limitation in the Discussion section.

“Is there a downfall to possibly eating too many nuts, (i.e. mold, pesticides), along these lines was there further qualification of the tree nuts to organic Yes/No.”

In the present study, it was not differentiated between organic and non-organic nuts.  The studies consulted did not specify this issue.

“Give above comments, the introduction and discussion should be reviewed”.

In the revised manuscript, the introduction, and discussion have been reviewed.

The revised manuscript has been edited by a native English speaker.

We hope that the manuscript is now acceptable for publication in Nutrients.

Yours sincerely,

Mª José Santi, MD, PhD.

Round 2

Reviewer 2 Report

Thanks for the consideration and further edits.

Modification of the title is more reflective of the data presented.

There are still concerns, however.  The main outcome is that the nuts are playing a role.  We know that the most robust study design is an RCT.  This could be suggested as a future direction for the work in the discussion, as there is still a chance that nuts have no association.

I do not recall if the issue of blindness was addressed, i.e. did the university students know that you were looking for an association with nut consumption and CRF.  If it was a general screening, this should be mentioned in the text.  Contrarily, if students did know you were looking for nut associations and CRF this also needs to be stated, as this certainly lends itself to an internal bias in the way the data was collected.  A statement should be added to the methods section.

A limitation section was added, what additional limitations are there in measuring BMI, where patients fasted, (semi-fasted), were they weighed at the same time of day, was the inter-rater reliability measured for the individual research assistants?  Was the repeat measure taken at the same time/date or at another time, if this was at a second time, when (2wk interval, 1mth?) this should be stated in the methods.

Are there additional limitations with the cardiorespiratory fitness?  These should be stated.

Line 52: before “against cardiometabolic diseases”…this does not read correctly, perhaps needs “to be active”, or “to be preventative”…not sure what the authors are trying to say here.

Line 59: should read “and are comprised of….”…

Line 61:  perhaps add another statement at the end of the sentence:  “therefore the benefits are believed to out way the negatives…” 

Lines 64-65:  each should have it’s own reference, citing one meta-analysis is not sufficient in this case, decrease fasting glucose (ref), total cholesterol (ref), LDL-cholesterol (ref), weight (ref), endothelial function (ref) and, gut microbiota modification (ref).

Lines:  42, 43, 46 vs. 81, 82, 87.  In some instances there is a space before the citation, in other instances there is not, would be good if consistent throughout the text.

Line 96, could a reference be added for the hypothesis…has “treat” food been negatively association with health outcomes, this is still vague, and should be cited.

Line 234:  no longer discussing nut consumption only, sofrito and fruits (Table 2 and Figure1?).  This discussion does not reflect that.

Were gender differences discussed or the lack thereof?

Author Response

March 5, 2020

Thank you for reading our manuscript titled “Association between a healthy dietary habit such as nuts consumption and cardiorespiratory fitness in young adults” (Manuscript ID: 719953) and reviewing it, which has helped us improve it. We revised the manuscript and all suggested changes have taken place.

Please, find below our point-by-point answers to the reviewer´s comments (blue color font). Changes made in the newly revised manuscript are in blue letters and the words deleted highlighted in yellow.

Reviewer # 2:

“Thanks for the consideration and further edits.

Modification of the title is more reflective of the data presented.

We thank the reviewer for these comments.

“There are still concerns, however. 

The main outcome is that the nuts are playing a role.  We know that the most robust study design is an RCT.  This could be suggested as a future direction for the work in the discussion, as there is still a chance that nuts have no association”.

In the revised manuscript, we have included it as a future direction for the work (Page 10, line 307-309).

“I do not recall if the issue of blindness was addressed, i.e. did the university students know that you were looking for an association with nut consumption and CRF.  If it was a general screening, this should be mentioned in the text.  Contrarily, if students did know you were looking for nut associations and CRF this also needs to be stated, as this certainly lends itself to an internal bias in the way the data was collected.  A statement should be added to the methods section”.

The students did not know the specific objectives of the work. They were informed that it was a general screening consisting of questionnaires, and anthropometric and CRF measurements.

In the revised manuscript, we have added a statement in the Methods section (Page 3, line 103-104). We also made it clear in the limitations section (Page 11, line 347-348).

 “A limitation section was added, what additional limitations are there in measuring BMI, where patients fasted, (semi-fasted), were they weighed at the same time of day, was the inter-rater reliability measured for the individual research assistants?  Was the repeat measure taken at the same time/date or at another time, if this was at a second time, when (2wk interval, 1mth?) this should be stated in the methods”.

In the revised manuscript, we have specified these issues in the Methods section (Page 3, line 134-135; page 4, line 136.140).

“Are there additional limitations with the cardiorespiratory fitness?  These should be stated”.

In the revised manuscript, we have added the limitations with the cardiorespiratory fitness in the limitations section (Page 11, line 353-355).

“Line 52: before “against cardiometabolic diseases”…this does not read correctly, perhaps needs “to be active”, or “to be preventative”…not sure what the authors are trying to say here”.

In the revised manuscript, we have replaced the word “against” by “in the prevention of” (Page 2, line 51).

“Line 59: should read “and are comprised of….”…”

In the new revised manuscript, we have corrected the sentence (Page 2, line 58).

“Line 61:  perhaps add another statement at the end of the sentence:  “therefore the benefits are believed to out way the negatives…” 

In the revised manuscript, we have added the sentence “Therefore, the benefits are believed to outweigh the advantages (Page 2, line 60-61).

“Lines 64-65:  each should have it’s own reference, citing one meta-analysis is not sufficient in this case, decrease fasting glucose (ref), total cholesterol (ref), LDL-cholesterol (ref), weight (ref), endothelial function (ref) and, gut microbiota modification (ref)”.

In the revised manuscript, we have added a reference to each variable. The references nº 9-14 (Page 2, line 64-66; page 13, line 416-437).

“Lines:  42, 43, 46 vs. 81, 82, 87.  In some instances there is a space before the citation, in other instances there is not, would be good if consistent throughout the text”.

In the revised manuscript, we have added a space before the citation throughout the text”.

“Line 96, could a reference be added for the hypothesis…has “treat” food been negatively association with health outcomes, this is still vague, and should be cited”.

In the revised manuscript, we have added a reference for the hypothesis (Page 3, line 96).

“Line 234:  no longer discussing nut consumption only, sofrito and fruits (Table 2 and Figure1?).  This discussion does not reflect that”.

In the revised manuscript, we have replaced “nut consumption”  by “specific MedD components” (Page 10, line 264).

“Were gender differences discussed or the lack thereof?”

In the revised manuscript, we have discussed the gender differences and the lack of differences (Page 10, line 268-283).

We hope that the manuscript is now acceptable for publication in Nutrients.

Yours sincerely,

Mª José Santi, MD, PhD.
